# Grounding Spatio-Temporal Language with Transformers

**Tristan Karch**[*]**, Laetitia Teodorescu**[*]
Inria - Flowers Team
Université de Bordeaux
`firstname.lastname@inria.fr`

**Katja Hofmann**
Microsoft Research
Cambridge, UK

**Clément Moulin-Frier**
Inria - Flowers team
Université de Bordeaux
ENSTA ParisTech

**Pierre-Yves Oudeyer**
Inria - Flowers team
Université de Bordeaux
ENSTA ParisTech

## Abstract

Language is an interface to the outside world. In order for embodied agents to use it, language must be grounded in other, sensorimotor modalities. While there is an extended literature studying how machines can learn grounded language, the topic of how to learn spatio-temporal linguistic concepts is still largely uncharted. To make progress in this direction, we here introduce a novel spatio-temporal language grounding task where the goal is to learn the meaning of spatio-temporal descriptions of behavioral traces of an embodied agent. This is achieved by training a truth function that predicts if a description matches a given history of observations. The descriptions involve time-extended predicates in past and present tense as well as spatio-temporal references to objects in the scene. To study the role of architectural biases in this task, we train several models including multimodal Transformer architectures; the latter implement different attention computations between words and objects across space and time. We test models on two classes of generalization: 1) generalization to randomly held-out sentences; 2) generalization to grammar primitives. We observe that maintaining object identity in the attention computation of our Transformers is instrumental to achieving good performance on generalization overall, and that summarizing object traces in a single token has little influence on performance. We then discuss how this opens new perspectives for language-guided autonomous embodied agents. We also release our code under open-source license as well as pretrained models and datasets to encourage the wider community to build upon and extend our work in the future.

## 1 Introduction

Building autonomous agents that learn to represent and use language is a long standing-goal in *Artificial Intelligence* [20, 39]. In developmental robotics [7], language is considered a cornerstone of development that enables embodied cognitive agents to learn more efficiently through social interactions with humans or through cooperation with other autonomous agents. It is also considered essential to develop complex sensorimotor skills, facilitating the representation of behaviors and actions.

*Embodied Language Grounding* [50] is the field that studies how agents can align language with their behaviors in order to extract the meaning of linguistic constructions. Early approaches in

---

[*]Equal Contribution

35th Conference on Neural Information Processing Systems (NeurIPS 2021).

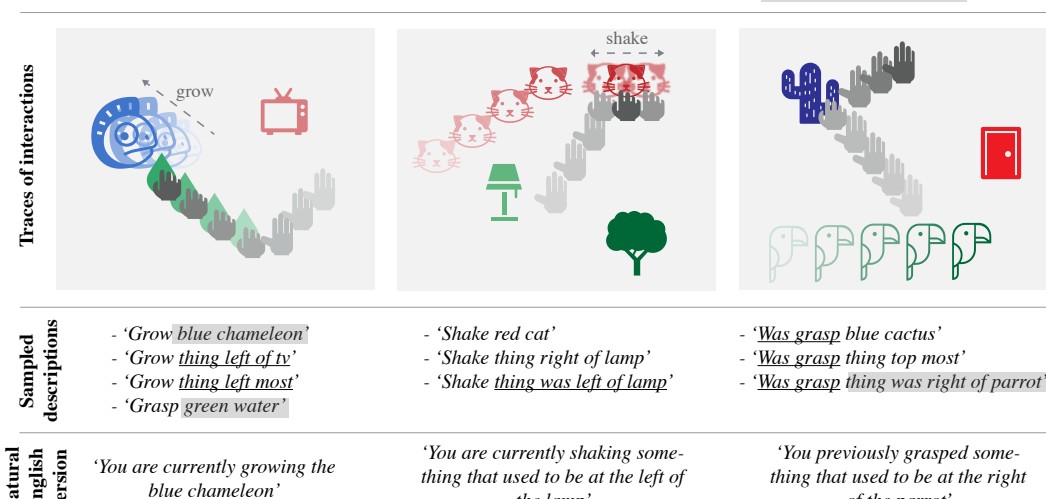

| (a) *Grow* action with spatial or attribute reference to object | (b) *Shake* action with spatio-temporal reference to object | (c) *Grasp* past actions description with spatio-temporal reference to object |
|---|---|---|

**Sampled descriptions**

| | | |
|---|---|---|
| - '*Grow* blue chameleon' | - '*Shake* red cat' | - '*Was grasp* blue cactus' |
| - '*Grow* thing left of tv' | - '*Shake* thing right of lamp' | - '*Was grasp* thing top most' |
| - '*Grow* thing left most' | - '*Shake* thing was left of lamp' | - '*Was grasp* thing was right of parrot' |
| - '*Grasp* green water' | | |

**Natural English version**

| | | |
|---|---|---|
| '*You are currently growing the blue chameleon*' | '*You are currently shaking something that used to be at the left of the lamp*' | '*You previously grasped something that used to be at the right of the parrot*' |

Figure 1: **Visual summary of the Temporal Playground environment:** At each episode (column a, b and c), the actions of an agent (represented by a hand) unfold in the environment and generate a trace of interactions between objects and the agent body. Given such a trace, the environment automatically generates a set of synthetic linguistic descriptions that are true at the end of the trace. In (a) the agent grows an object which is described with spatial (underlined) or attribute (highlighted) reference. In (b) it shakes an object which is described with attribute, spatial or spatio-temporal (underlined) reference. In (c) it has grasped an object (past action underlined) which is described with attribute, spatial or spatio-temporal (highlighted) reference. The natural english version is given for illustrative purposes.

developmental robotics studied how various machine learning techniques, ranging from neural networks [40, 45, 24] to non-negative matrix factorization [32], could enable the acquisition of grounded compositional language [42, 41]. This line of work was recently extended using techniques for *Language conditioned Deep Reinforcement Learning* [31]. Among these works we can distinguish mainly three language grounding strategies. The first one consists of directly grounding language in the behavior of agents by training goal-conditioned policies satisfying linguistic instructions [40, 45, 23, 22, 9]. The second aims at extracting the meaning of sentences from mental simulations (i.e. generative models) of possible sensorimotor configurations matching linguistic descriptions [32, 1, 12, 34]. The third strategy searches to learn the meaning of linguistic constructs in terms of outcomes that agents can observe in the environment. This is achieved by training a truth function that detects if descriptions provided by an expert match certain world configurations. This truth function can be obtained via *Inverse Reinforcement Learning* [49, 3] or by training a multi-modal binary classifier [14]. Previous work [14, 3] has shown that access to this reward is enough for sucessfully grounding language in instruction-following agents.

While all the above-mentioned approaches consider language that describes immediate and instantaneous actions, we argue that it is also important for agents to grasp language expressing concepts that are relational and that span multiple time steps. We thus propose to study the grounding of new spatio-temporal concepts enabling agents to ground time extended predicates (Fig. 1a) with complex spatio-temporal references to objects (Fig. 1b) and understand both present and past tenses (Fig. 1c). To do so we choose the third strategy mentioned above, i.e. to train a truth function that predicts when descriptions match traces of experience. This choice is motivated by two important considerations. First, prior work showed that learning truth functions was key to foster generalization [3], enabling agents to efficiently self-train policies via goal imagination [14] and goal relabeling [12]. Hence the truth function is an important and self-contained component of larger learning systems. Second, this strategy allows to carefully control the distribution of experiences and descriptions perceived by the agent.

The Embodied Language Grounding problem has a relational structure. We understand the meaning of words by analyzing the relations they state in the world [18]. The relationality of spatial and temporal concepts has long been identified in the field of pragmatics [43, 44] (see Supplementary Section C for additional discussion). Furthermore actions themselves are relations between subjects and objects, and can be defined in terms of affordances of the agent [19]. We acknowledge this and provide the right relational inductive bias [5] to our architectures through the use of Transformers [46]. We propose a formalism unifying three variants of a multi-modal transformer inspired by Ding et al. [16] that implement different relational operations through the use of hierarchical attention. We measure the generalization capabilities of these architectures along three axes 1) generalization to new traces of experience; 2) generalization to randomly held out sentences; 3) generalization to grammar primitives, systematically held out from the training set as in Ruis et al. [37]. We observe that maintaining object identity in the attention computation of our Transformers is instrumental to achieving good performance on generalization overall. We also identify specific relational operations that are key to generalize on certain grammar primitives.

**Contributions.**    This paper introduces:

1. A new Embodied Language Grounding task focusing on spatio-temporal language;
2. A formalism unifying different relational architectures based on Transformers expressed as a function of mapping and aggregation operations;
3. A systematic study of the generalization capabilities of these architectures and the identification of key components for their success on this task.

## 2    Methods

### 2.1    Problem Definition

We consider the setting of an embodied agent behaving in an environment. This agent interacts with the surrounding objects over time, during an episode of fixed length ($T$). Once this episode is over, an oracle provides exhaustive feedback in a synthetic language about everything that has happened. This language describes actions of the agent over the objects and includes spatial and temporal concepts. The spatial concepts are reference to an object through its spatial relation with others (Fig. 1a), and the temporal concepts are the past modality for the actions of the agent (Fig. 1c), past modality for spatial relations (Fig. 1b), and actions that unfold over time intervals. The histories of states of the agent's body and of the objects over the episode as well as the associated sentences are recorded in a buffer $\mathcal{B}$. From this setting, and echoing previous work on training agents from descriptions, we frame the Embodied Language Grounding problem as learning a parametrized truth function $R_\theta$ over couples of observations traces and sentences, tasked with predicting whether a given sentence $W$ is true of a given episode history $S$ or not. Formally, we aim to minimize:

$$\mathbb{E}_{(S,W)\sim\mathcal{B}}\big[\mathcal{L}(R_\theta(S,W), r(S,W))\big]$$

where $\mathcal{L}$ denotes the cross-entropy loss and $r$ denotes the ground truth boolean value for sentence $W$ about trace $S$.

### 2.2    Temporal Playground

In the absence of any dedicated dataset providing spatio-temporal descriptions from behavioral traces of an agent, we introduce *Temporal Playground* (Fig. 1) an environment coupled with a templated grammar designed to study spatio-temporal language grounding. The environment is a 2D world, with procedurally-generated scene containing $N = 3$ objects sampled from 32 different object types belonging to 5 categories. Each object has a continuous 2D position, a size, a continuous color code specified by a 3D vector in RGB space, a type specified by a one-hot vector, and a boolean unit specifying whether it is grasped. The size of the object feature vector ($o$) is $|o| = 39$. The agent's body has its 2D position in the environment and its gripper state (grasping or non-grasping) as features (body feature vector ($b$) of size $|b| = 3$). In this environment, the agent can perform various actions over the length ($T$) of an episode. Some of the objects (the animals) can move independently. Objects can also interact: if the agent brings food or water to an animal, it will grow in size; similarly, if water is brought to a plant, it will grow. At the end of an episode, a generative grammar generates

sentences describing all the interactions that occurred. A complete specification of the environment as well as the BNF of the grammar can be found in Supplementary Section A.2.

**Synthetic language.** To enable a controlled and systematic study of how different types of spatio-temporal linguistic meanings can be learned, we argue it is necessary to first conduct a systematic study with a controlled synthetic grammar. We thus consider a synthetic language with a vocabulary of size $53$ and sentences with a maximum length of $8$. This synthetic language facilitates the generation of descriptions matching behavioral traces of the agent. Moreover, it allows us to express four categories of concepts associated with specific words. Thus, the generated sentences consist in four conceptual types based on the words they involve:

- **Sentences involving basic concepts.** This category of sentences talk about present-time events by referring to objects and their attributes. Sentences begin with the *'grasp'* token combined with any object. Objects can be named after their category (eg. *'animal', 'thing'*) or directly by their type (*'dog', 'door', 'algae', etc.*). Finally, the color (*'red','blue','green'*) of objects can also be specified.
- **Sentences involving spatial concepts.** This category of sentences additionally involve one-to-one spatial relations and one-to-all spatial relations to refer to objects. An object can be *'left of'* another object (reference is made in relation to a single other object), or can be the *'top most'* object (reference is made in relation with all other objects). Example sentences include *'grasp thing bottom of cat'* or *'grasp thing right most'*.
- **Sentences involving temporal concepts**. This category of sentences involves talking about temporally-extended predicates and the past tense, without any spatial relations. The two temporal predicates are denoted with the words *'grow'* and *'shake'*. The truth value of these predicates can only be decided by looking at the temporal evolution of the object's size and position respectively. A predicate is transposed at the past tense if the action it describes was true at some point in the past and is no longer true in the present, this is indicated by adding the modifier *'was'* before the predicate. Example sentences include *'was grasp red chameleon'* (indicating that the agent grasped the red chameleon and then released it) and *'shake bush'*;
- **Sentences involving spatio-temporal concepts**. Finally, we consider the broad class of spatio-temporal sentences that combine spatial reference and temporal or past-tense predicates. These are sentences that involve both the spatial and temporal concepts defined above. Additionally, there is a case of where the spatial and the temporal aspects are entangled: past spatial reference. This happens when an object is referred to by its previous spatial relationship with another object. Consider the case of an animal that was at first on the bottom of a table, then moved on top, and then is grasped. In this case we could refer to this animal as something that was previously on the bottom of the table. We use the same *'was'* modifier as for the past tense predicates; and thus we would describe the action as *'Grasp thing was bottom of table'*.

## 2.3 Architectures

In this section we describe the architectures used as well as their inputs. Let one input sample to our model be $I = (S, W)$, where $(S_{i,t})_{i,t}$ represents the objects' and body's evolution, and $(W_l)_l$ represents the linguistic observations. $S$ has a spatial (or entity) dimension indexed by $i \in [0..N]$ and a temporal dimension indexed by $t \in [1..T]$; for any $i, t$, $S_{i,t}$ is a vector of observational features. Note that by convention, the trace $(S_{0,t})_t$ represents the body's features, and the traces $(S_{i,t})_{t,i>0}$ represents the other objects' features. $W$ is a 2-dimensional tensor indexed by the sequence $l \in [1..L]$; for any $l$, $W_l \in \mathbb{R}^{d_W}$ is a one-hot vector defining the word in the dictionary. The output to our models is a single scalar between $0$ and $1$ representing the probability that the sentence encoded by $W$ is true in the observation trace $S$.

**Transformer Architectures.** To systematically study the influence of architectural choices on language performance and generalization in our spatio-temporal grounded language context, we define a set of mapping and aggregation operations that allows us to succinctly describe different models in a unified framework. We define:

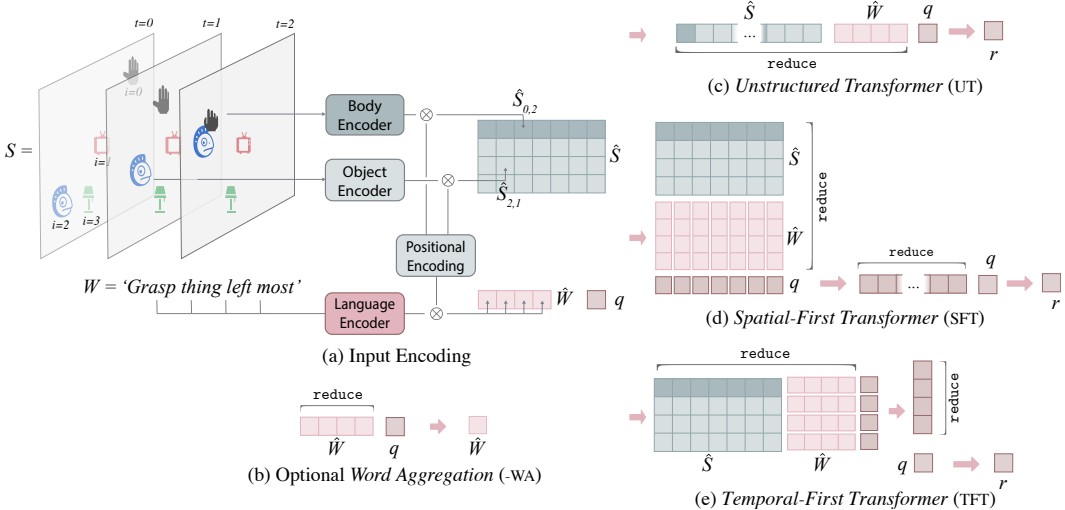

(a) Input Encoding

(b) Optional *Word Aggregation* (-WA)

(c) *Unstructured Transformer* (UT)

(d) *Spatial-First Transformer* (SFT)

(e) *Temporal-First Transformer* (TFT)

Figure 2: **Visual summary of the architectures used.** We show the details of UT, SFT and TFT respectively in subfigures (c), (d), (e), as well as a schematic illustration of the preprocessing phase (a) and the optional word-aggregation procedure (b).

- An aggregation operation based on a Transformer model, called `reduce`. `reduce` is a parametrized function that takes 3 inputs: a tensor, a dimension tuple $D$ over which to reduce and a query tensor (that has to have the size of the reduced tensor). $R$ layers of a Transformer are applied to the input-query concatenation and are then queried at the position corresponding to the query tokens. This produces an output reduced over the dimensions $D$.
- A casting operation called `cast`. `cast` takes as input 2 tensors $A$ and $B$ and a dimension $d$. $A$ is flattened, expanded so as to fit the tensor $B$ in all dimensions except $d$, and concatenated along the $d$ dimension.
- A helper expand operation called `expand` that takes as arguments a tensor and an integer $n$ and repeats the tensor $n$ times.

Using those operations, we define three architectures: one with no particular bias (*Unstructured Transformer*, inspired by Ding et al. [16], or UT); one with a spatial-first structural bias – objects and words are aggregated along the spatial dimension first (*Spatial-First Transformer* or SFT); and one with a temporal-first structural bias – objects and words are aggregated along the temporal dimension first (*Temporal-First Transformer*, or TFT).

Before inputting the observations of bodies and objects $S$ and the language $W$ into any of the Transformer architectures, they are projected to a common dimension (see Supplementary Section B.2 for more details). A positional encoding [46] is then added along the time dimension for observations and along the sequence dimension for language; and finally a one-hot vector indicating whether the vector is observational or linguistic is appended at the end. This produces the modified observation-language tuple $(\hat{S}, \hat{W})$. We let:

$$\text{UT}(\hat{S}, \hat{W}) := \texttt{reduce}(\texttt{cast}(\hat{S}, \hat{W}, 0), 0, q)$$

$$\text{SFT}(\hat{S}, \hat{W}, q) := \texttt{reduce}(\texttt{reduce}(\texttt{cast}(\hat{W}, \hat{S}, 0), 0, \texttt{expand}(q, T)), 0, q)$$

$$\text{TFT}(\hat{S}, \hat{W}, q) := \texttt{reduce}(\texttt{reduce}(\texttt{cast}(\hat{W}, \hat{S}, 1), 1, \texttt{expand}(q, N+1)), 0, q)$$

where $T$ is the number of time steps, $N$ is the number of objects and $q$ is a learned query token. See Fig. 2 for an illustration of these architectures.

Note that SFT and TFT are transpose versions of each other: SFT is performing aggregation over space first and then time, and the reverse is true for TFT. Additionally, we define a variant of each of these architectures where the words are aggregated before being related with the observations. We name these variants by appendding -WA (word-aggregation) to the name of the model (see Fig. 2 (b)).

$$\hat{W} \leftarrow \texttt{reduce}(\hat{W}, 0, q)$$

We examine these variants to study the effect of letting word-tokens directly interact with object-token through the self-attention layers vs simply aggregating all language tokens in a single embedding and letting this vector condition the processing of observations. The latter is commonly done in the language-conditioned RL and language grounding literature [11, 3, 25, 37], using the language embedding in FiLM layers [36] for instance. Finding a significant effect here would encourage using architectures which allow direct interactions between the word tokens and the objects they refer to.

**LSTM Baselines.** We also compare some LSTM-based baselines on this task; their architecture is described in more detail in Supplementary Section B.3.

## 2.4 Data Generation, Training and Testing Procedures

We use a bot to generate the episodes we train on. The data collected consists of 56837 trajectories of $T = 30$ time steps. Among the traces some descriptions are less frequent than others but we make sure to have at least 50 traces representing each of the 2672 descriptions we consider. We record the observed episodes and sentences in a buffer, and when training a model we sample $(S, W, r)$ tuples with one observation coupled with either a true sentence from the buffer or another false sentence generated from the grammar. More details about the data generation can be found in Supplementary Section B.1.

For each of the Transformer variants (6 models) and the LSTM baselines (2 models) we perform an hyper parameter search using 3 seeds in order to extract the best configuration. We extract the best condition for each model by measuring the mean $F_1$ on a testing set made of uniformly sampled descriptions from each of the categories define in section 2.2. We use the $F_1$ score because testing sets are imbalanced (the number of traces fulfilling each description is low). We then retrain best configurations over 10 seeds and report the mean and standard deviation (reported as solid black lines in Fig. 3 and Fig. 4) of the averaged $F_1$ score computed on each set of sentences. When statistical significance is reported in the text, it is systematically computed using a two-tail Welch's t-test with null hypothesis $\mu_1 = \mu_2$, at level $\alpha = 0.05$ [13]. Details about the training procedure and the hyper parameter search are provided in Supplementary Section B.4.

# 3 Experiments and Results

## 3.1 Generalization abilities of models on non-systematic split by categories of meaning

In this experiment, we perform a study of generalization to new sentences from known observations. We divide our set of test sentences in four categories based on the categories of meanings listed in Section 2.2: Basic, Spatial, Spatio-Temporal and Temporal. We remove 15% of all possible sentences in each category from the train set and evaluate the F1 score on those sentences. The results are provided in Fig. 3.

First, we notice that over all categories of meanings, all UT and TFT models, with or without word-aggregation, perform extremely well compared to the LSTM baselines, with all these four models achieving near-perfect test performance on the Basic sentences, with very little variability across the 10 seeds. We then notice that all SFT variants perform poorly on all test categories, in line or worse than the baselines. This is particularly visible on the spatio-temporal category, where the SFT models perform at $0.75 \pm 0.020$ whereas the baselines perform at $0.80 \pm 0.019$ . This suggests that across tasks, it is harmful to aggregate each scene plus the language information into a single vector. This may be due to the fact that objects lose their identity in this process, since information about all the objects becomes encoded in the same vector. This may make it difficult for the network to perform computations about the truth value of predicate on a single object.

Secondly, we notice that the word-aggregation condition seems to have little effect on the performance on all three Transformer models. We only observe a significant effect for UT models on spatio-temporal concepts (p-value = 2.38e-10). This suggests that the meaning of sentences can be adequately summarised by a single vector; while maintaining separated representations for each object is important for achieving good performance it seems unnecessary to do the same for linguistic input. However we notice during our hyperparameter search that our -WA models are not very robust to hyperparameter choice, with bigger variants more sensitive to the learning rate.

Thirdly, we observe that for our best-performing models, the basic categories of meanings are the easiest, with a mean score of $1.0 \pm 0.003$ across all UT and TFT models, then the spatial ones at $0.96 \pm 0.020$, then the temporal ones at $0.96 \pm 0.009$, and finally the spatio-temporal ones at $0.89 \pm 0.027$. This effectively suggests, as we hypothesised, that sentences containing spatial relations or temporal concepts are harder to ground than those who do not.

**Known sentences with novel observations.** We also examine the mean performance of our models for sentences in the training set but evaluated on a set of *new observations*: we generate a new set of rollouts on the environment, and only evaluate the model on sentences seen at train time (plots are reported in Supplementary Section D). We see the performance is slightly better in this case, especially for the LSTM baselines ($0.82 \pm 0.031$ versus $0.79 \pm 0.032$), but the results are comparable in both cases, suggesting that the main difficulty for models lies in grounding spatio-temporal meanings and not in linguistic generalization for the type of generalization considered in this section.

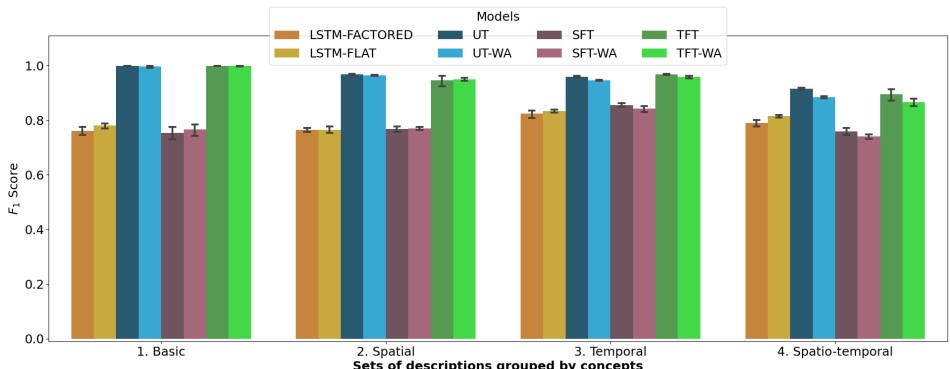

Figure 3: **F1 scores for all the models on randomly held-out sentences.** $F_1$ is measured on separated sets representing each category of concepts defined in Section 2.2.

## 3.2  Systematic generalization on withheld combinations of words

In addition to the previous generalization studies, we perform an experiment in a harder linguistic generalization setting where we systematically remove binary combinations in our train set. This is in line with previous work on systematic generalization on deep learning models [29, 37, 26]. We create five test sets to examine the abilities of our models to generalize on binary combinations of words that have been systematically removed from the set of training sentences, but whose components have been seen before in other contexts. Our splits can be described by the set of forbidden combinations of words as:

1. **Forbidden object-attribute combinations.** remove from the train set all sentences containing *'red cat'*, *'blue door'* and *'green cactus'*. This tests the ability of models to recombine known objects with known attributes;
2. **Forbidden predicate-object combination.** remove all sentences containing *'grow'* and all objects from the *'plant'* category. This tests the model's ability to apply a known predicate to a known object in a new combination;
3. **Forbidden one-to-one relation.** remove all sentences containing *'right of'*. Since the *'right'* token is already seen as-is in the context of one-to-all relations (*'right most'*), and other one-to-one relations are observed during training, this tests the abilities of models to recombine known directions with in a known template;
4. **Forbidden past spatial relation.** remove all sentences containing the contiguous tokens *'was left of'*. This tests the abilities of models to transfer a known relation to the past modality, knowing other spatial relations in the past;
5. **Forbidden past predicate.** remove all sentences containing the contiguous tokens *'was grasp'*. This tests the ability of the model to transfer a known predicate to the past modality, knowing that it has already been trained on other past-tense predicates.

To avoid retraining all models for each split, we create one single train set with all forbidden sentences removed and we test separately on all splits. We use the same hyperparameters for all models than in the previous experiments. The results are reported in Fig. 4.

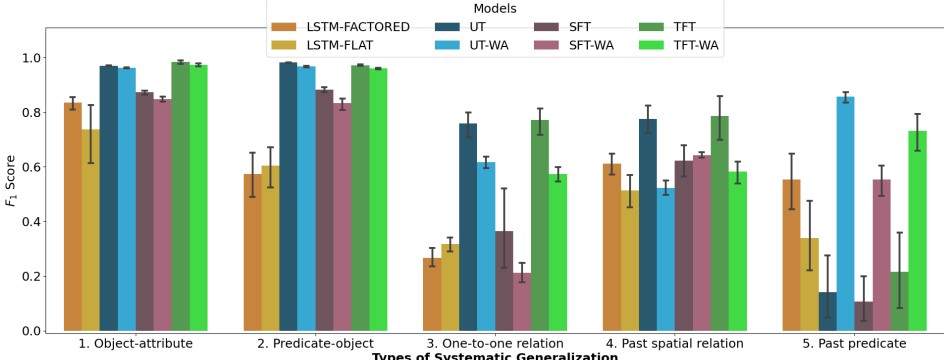

Figure 4: **$F_1$ scores of all the models on systematic generalization splits.** $F_1$ is measured on separated sets representing each of the forbidden combinations of word defined above.

First we can notice that the good test scores obtained by the UT and TFT models on the previous sections are confirmed in on this experiment: they are the best performing models overall. We then notice that the first two splits, corresponding to new attribute-object and predicate-object combinations, are solved by the UT and TFT models, while the SFT models and the LSTM baselines struggle to achieve high scores. For the next 3 splits, which imply new spatial and temporal combinations, the scores overall drop significantly; we also observe much wider variability between seeds for each model, perhaps suggesting the various strategies adopted by the models to fit the train set have very different implications in terms of systematic generalization on spatial and temporal concepts. This very high variability between seeds on systematic generalization scores are reminiscent of the results obtained on the gSCAN benchmark [37].

Additionally, for split 3, which implies combining known tokens to form a new spatial relation, we observe a significant drop in generalization for the word-aggregation (WA) conditions, consistent across models (on average across seeds, $-0.14 \pm 0.093$, $-0.15 \pm 0.234$ and $-0.20 \pm 0.061$ for UT, SFT and TFT resp. with p-values $< 1$e-04 for UT and SFT). This may be due to the fact that recombining any one-to-one relation with the known token *right* seen in the context of one-to-all relations requires a separate representation for each of the linguistic tokens. The same significant drop in performance for the WA condition can be observed for UT and TFT in split 4, which implies transferring a known spatial relation to the past.

However, very surprisingly, for split 5 – which implies transposing the known predicate *grasp* to the past tense – we observe a very strong effect in the opposite direction: the WA condition seems to help generalizing to this unknown past predicate (from close-to-zero scores for all transformer models, the *WA* adds on average $0.71 \pm 0.186$, $0.45 \pm 0.178$ and $0.52 \pm 0.183$ points for UT, ST and TT resp. and p-values $< 1$e-05). This may be due to the fact that models without WA learn a direct and systematic relationship between the *grasp* token and grasped objects, as indicated in their features; this relation is not modulated by the addition of the *was* modifier as a prefix to the sentence. Models do not exhibit the same behavior on split 4, which has similar structure (transfer the relation *left of* to the past). This may be due to the lack of variability in instantaneous predicates (only the *grasp* predicate); whereas there are several spatial relations (4 one-to-one, 4 one-to-all).

**Control experiment.** We evaluate previously trained models on a test set containing hard negative examples. The aim of this experiment is to ensure that models truly identify the compositional structure of our spatio-temporal concepts and do not simply perform unit concept recognition. We select negative pairs (trajectory, description) so that the trajectories contain either the object or the action described in the positive example. Results are provided in Fig. 5. We observe a slight decrease of performances on all 5 categories (drop is less than 5%), demonstrating that the models do in fact represent the meaning of the sentence and not simply the presence or absence of a particular object or predicate.

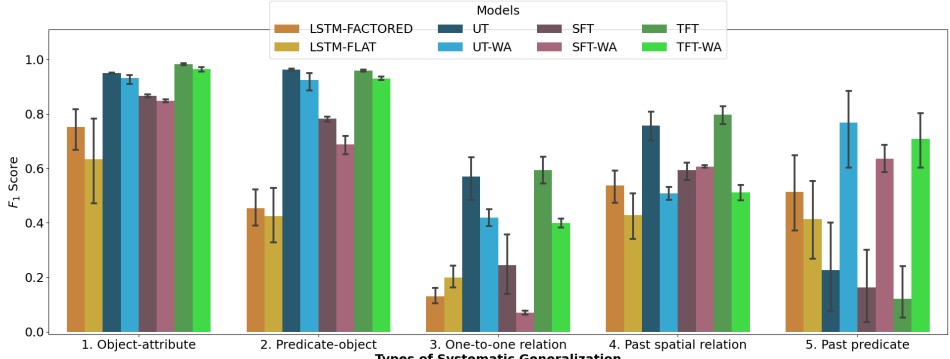

Figure 5: **Control experiment: F1** scores for all the models on systematic generalization splits in the hard negative examples setting.

# 4 Related Work

The idea that agents should learn to represent and ground language in their experience of the world has a long history in developmental robotics [50, 39, 40, 7] and was recently extended in the context of Language Conditioned Deep Reinforcement Learning [11, 22, 31, 3]. These recent approaches often consider navigation [10, 9] or object manipulation [1, 22] tasks and are always using instructive language. Meanings typically refer to instantaneous actions and rarely consider spatial reference to objects [35]. Although our environment includes object manipulations, we here tackle novel categories of meanings involving the grounding of spatio-temporal concepts such as the past modality or complex spatio-temporal reference to objects.

We evaluate our learning architectures on their ability to generalise to sets of descriptions that contain systematic differences with the training data so as to assess whether they correctly model grammar primitives. This procedure is similar to the *gSCAN* benchmark [37]. This kind of compositional generalisation is referred as 'systematicity' by Hupkes et al. [26]. Environmental drivers that facilitate systematic generalization are also studied by Hill et al. [23]. Although Hupkes et al. [26] consider relational models in their work, they do not evaluate their performance on a *Language Grounding* task. Ruis et al. [37] consider an Embodied Language Grounding setup involving one form of time-extended meanings (adverbs), but do not consider the past modality and spatio-temporal reference to objects, and do not consider learning truth functions. Also, they do not consider learning architectures that process sequences of sensorimotor observations. To our knowledge, no previous work has conducted systematic generalization studies on an Embodied Language Grounding task involving spatio-temporal language with Transformers.

The idea that relational architectures are relevant models for Language Grounding has been previously explored in the context of *Visual Reasoning*. They were indeed successfully applied for spatial reasoning in the visual question answering task *CLEVR* [38]. With the recent publication of the video reasoning dataset *CLEVRER* [47], those models were extended and demonstrated abilities to reason over spatio-temporal concepts, correctly answering causal, predictive and counterfactual questions [16]. In contrast to our study, these works around CLEVRER do not aim to analyze spatio-temporal language and therefore do not consider time-extended predicates or spatio-temporal reference to objects in their language, and do not study properties of systematic generalization over sets of new sentences.

# 5 Discussion and Conclusion

In this work, we have presented a first step towards learning Embodied Language Grounding of spatio-temporal concepts, framed as the problem of learning a truth function that can predict if a given sentence is true of temporally-extended observations of an agent interacting with a collection of objects. We have studied the impact of architectural choices on successful grounding of our artificial spatio-temporal language. We have modelled different possible choices for aggregation of

observations and language as hierarchical Transformer architectures. We have demonstrated that in our setting, it is beneficial to process temporally-extended observations and language tokens side-by-side, as evidenced by the good score of our Unstructured Transformer variant. However, there seems to be only minimal effect on performance in aggregating temporal observations along the temporal dimension first – compared to processing all traces and the language in an unstructured manner – as long as object identity is preserved. This can inform architectural design in cases where longer episode lengths make it impossible to store all individual timesteps for each object; our experiments provide evidence that a temporal summary can be used in these cases. Our experiments with systematic dimensions of generalization provide mixed evidence for the influence of summarizing individual words into a single vector, showing it can be detrimental to generalize to novel word combinations but also can help prevent overgeneralization of a relation between a single word and a single object without considering the surrounding linguistic context.

**Limitations and further work.** There are several limitations of our setup which open important opportunities for further work. First, we have used a synthetic language that could be extended: for instance with more spatial relations and relations that are more than binary. Another axis for further research is using low-level observations. In our setting, we wanted to disentangle the effect of structural biases on learning spatio-temporal language from the problem of extracting objects from low level observations [6, 21, 17, 30, 8] in a consistent manner over time (object permanence [15, 48]). Further steps in this direction are needed, and it could allow us to define richer attributes (related to material or texture) and richer temporal predicates (such as breaking, floating, etc). Finally, we use a synthetic language which is far from the richness of the natural language used by humans, but previous work has shown that natural language can be projected onto the subspace defined by synthetic language using the semantic embeddings learned by large language models [33]: this opens up be a fruitful avenue for further investigation.

A further interesting avenue for future work would be to use the grounding provided by this reward function to allow autonomous language-conditioned agents to target their own goals [14]. In this sense, the truth function can be seen as a goal-achievement function or reward function. While generalization performance of our method is not perfect, the good overall f1 scores of our architectures imply that they can be directly transferred to a more complete RL setup to provide signal for policies conditioned on spatio-temporal language.

**Broader Impact.** This work provides a step in the direction of building agents that better understand how language relates to the physical world; this can lead to personal robots that can better suit the needs of their owners because they can be interacted with using language. If successfully implemented, this technology can raise issues concerning automation of certain tasks resulting in loss of jobs for less-qualified workers.

**Links.** The source code as well as the generated datasets can be found at `https://github.com/flowersteam/spatio-temporal-language-transformers`.

## Acknowledgments and Disclosure of Funding

Tristan Karch is partly funded by the French Ministère des Armées - Direction Générale de l'Armement. Laetitia Teodorescu is supported by Microsoft Research through its PhD Scholarship Programme. This work was performed using HPC resources from GENCI-IDRIS (Grant 2020-A0091011996)

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
