SUPPLEMENTARY MATERIAL

# A  Supplementary: Temporal Playground Specifications

## A.1  Environment

Temporal Playground is a procedurally generated environment consisting of 3 objects and an agent's body. There are 32 types of objects, listed in Fig. 6 along with 5 object categories. Each object has a continuous 2D position, a size, a continuous color code specified by a 3D vector in RGB space, a type specified by a one-hot vector, and a boolean unit specifying whether it is grasped. Note that categories are not encoded in the objects' features. The agent's body has its 2D position in the environment and its gripper state (grasping or non-grasping) as features. The size of the body feature vector is 3 while the object feature vector has a size of 39. This environment is a spatio-temporal extension of the one used in this work [14].

All positions are constrained within $[-1, 1]^2$. The initial position of the agent is $(0, 0)$ while the initial object positions are randomized so that they are not in contact $(d(obj_1, obj_2) > 0.3)$. Object sizes are sampled uniformly in $[0.2, 0.3]$, the size of the agent is $0.05$. Objects can be grasped when the agent has nothing in hand, when it is close enough to the object center $(d(\text{agent}, obj) < (size(\text{agent}) + size(obj))/2)$ and the gripper is closed $(1, -1$ when open). When a supply is on an animal or water is on a plant (contact define as distance between object being equal to the mean size of the two objects $d = (size(obj_1) + size(obj_2))/2)$, the object will grow over time with a constant growth rate until it reaches the maximum size allowed for objects or until contact is lost.

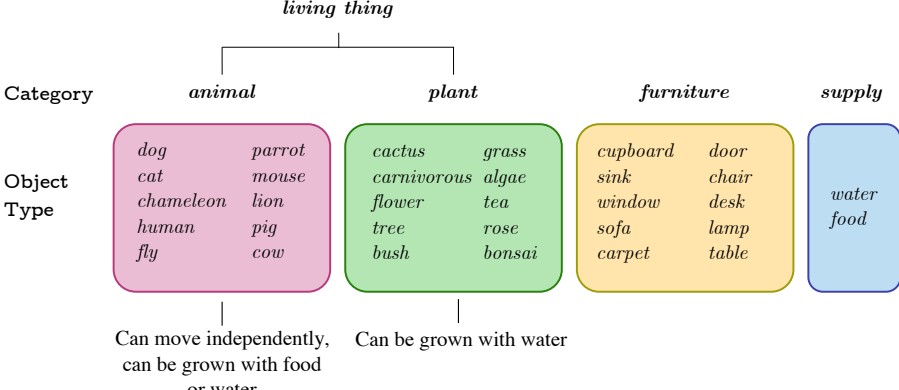

Figure 6: **Representation of possible objects types and categories**. Information about the possible interactions between objects are also given.

## A.2 Language

**Grammar.** The synthetic language we use can be decomposed into two components: the instantaneous grammar and the temporal logic. Both are specified through the BNF given in Figure 7.

Instantaneous grammar:

```
             ::= <pred> <thing_A>
<pred>          ::= grow | grasp | shake
<thing_A>       ::= <thing_B> | <attr> <thing_B> | thing <localizer> |
                    thing <localizer_all>
<localizer>     ::= left of <thing_B> | right of <thing_B> |
                    bottom of <thing_B> | top of <thing_B>
<localizer_all> ::= left most | right most | bottom most | top most
<thing_B>       ::= dog | cat | … | thing
<attr>          ::= blue | green | red
```

Temporal aspect:

```
             ::= was <pred> <thing_A>
<thing_A>       ::= thing was <localizer> | thing was <localizer_all>
```

Figure 7: **BNF of the grammar used in Temporal Playground**. The instantaneous grammar allows generating true sentences about predicates, spatial relations (one-to-one and one to all). These sentences are then processed by the temporal logic to produce the linguistic descriptions of our observations; this step is illustrated in the Temporal Aspect rules. See the main text for information on how these sentences are generated.

**Concept Definition.** We split the set of all possible descriptions output by our grammar into four conceptual categories according to the rules given in Table 1.

| Concept | BNF | Size |
|---|---|---|
| **1. Basic** | `     ::=  <pred> <thing_A>`
`<pred>   ::=  grasp`
`<thing_A> ::=  <thing_B> | <attr> <thing_B` | 152 |
| **2. Spatial** | `     ::=  <pred> <thing_A>`
`<pred>   ::=  grasp`
`<thing_A> ::=  <thing <localizer> | thing <localizer_all>` | 156 |
| **3. Temporal** | `     ::=  <pred_A> <thing_A> | was <pred_B> <thing_A>`
`<pred_A> ::=  grow | shake`
`<pred_B> ::=  grasp | grow | shake`
`<thing_A> ::=  <thing_B> | <attr> <thing_B>` | 648 |
| **4. Spatio-Temporal** | `     ::=  <pred_A> <thing_A> | was <pred_B> <thing_A>`
`             <pred_C> <thing_C>`
`<pred_A> ::=  grow | shake`
`<pred_B> ::=  grasp | grow | shake`
`<pred_C> ::=  grasp`
`<thing_A> ::=  thing <localizer> | thing <localizer_all> |`
`             thing was <localizer> |`
`             thing was <localizer_all> |`
`<thing_C> ::=  thing was <localizer> |`
`             thing was <localizer_all>` | 1716 |

Table 1: **Concept categories with their associated BNF.** `<thing_B>`, `<attr>`, `<localizer>` and `<localizer_all>` are given in Fig. 7.

# B Supplementary Methods

## B.1 Data Generation

**Scripted bot.** To generate the traces matching the descriptions of our grammar we define a set of scenarii that correspond to sequences of actions required to fulfill the predicates of our grammar, namely *grasp*, *grow* and *shake*. Those scenarii are then conditioned on a boolean that modulates them to obtain a mix of predicates in the present and the past tenses. For instance, if a *grasp* scenario is sampled, there will be a 50% chance that the scenario will end with the object being grasped, leading to a present-tense description; and a 50% chance that the agent releases the object, yielding a past tense description.

**Description generation from behavioral traces of the agent.** For each time step, the instantaneous grammar generates the set of all true instantaneous sentences using a set of filtering operations similar to the one used in CLEVR [27], without the past predicates and past spatial relations. Then the temporal logic component uses these linguistic traces in the following way: if a given sentence for a predicate is true in a past time step and false in the present time step, the prefix token 'was' is prepended to the sentence; similarly, if a given spatial relation is observed in a previous time step and unobserved in the present, the prefix token 'was' is prepended to the spatial relation.

## B.2 Input Encoding

We present the input processing in Fig. 8. At each time step $t$, the body feature vector $b_t$ and the object features vector $o_{i,t}$, $i = 1, 2, 3$ are encoded using two single-layer neural networks whose output are of size $h$. Similarly, each of the words of the sentence describing the trace (represented as one-hot vectors) is encoded and projected in the dimension of size $h$. We concatenate to the vector obtained a modality token $m$ that defines if the output belongs to the scene $(1, 0)$ or to the description $(0, 1)$. We then feed the resulting vectors to a positional encoding that modulates the vectors according to the time step in the trace for $b_t$ and $o_{i,t}$, $i = 1, 2, 3$ and according to the position of the word in the description for $w_l$.

We call the encoded body features $\hat{b}_t$ and it corresponds to $\hat{S}_{0,t}$ of the input tensor of our model (see Fig. 2 in the Main document). Similarly, $\hat{o}_{i,t}$, $i = 1, 2, 3$ are the encoded object features corresponding to $\hat{S}_{i,t}$, $i = 1, 2, 3$. Finally $\hat{w}_l$ are the encoded words and the components of tensor $\hat{W}$.

We call $h$ the hidden size of our models and recall that $|\hat{b}_t| = |\hat{o}_{i,t}| = |\hat{w}_l| = h + 2$. This parameter is varied during the hyper-parameter search.

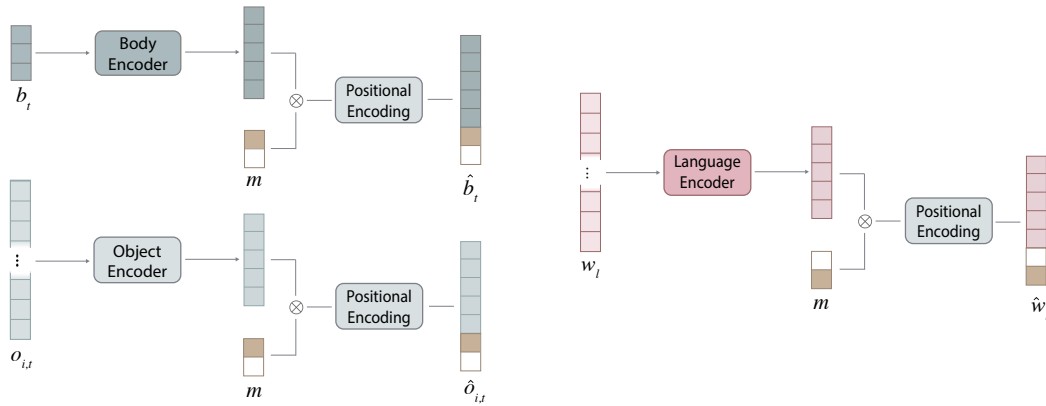

Figure 8: **Input encoding.** Body, words and objects are all projected in the same dimension.

## B.3 Details on LSTM models

To provide baseline models on our tasks we consider two LSTM variants. They are interesting baselines because they do not perform any relational computation except for relations between inputs at successive time steps. We consider the inputs as they were defined in Section 2.3 of the main paper. We consider two LSTM variants:

1. LSTM-FLAT: This variant has two internal LSTM: one that processes the language and one that processes the scenes as concatenations of all the body and object features. This produces two vectors that are concatenated into one, which is then run through an MLP and a final softmax to produce the final output.

2. LSTM-FACTORED: This variant independently processes the different body and object traces, which have previously been projected to the same dimension using a separate linear projection for the object and for the body. The language is processed by a separate LSTM. These body, object and language vectors are finally concatenated and fed to a final MLP and a softmax to produce the output.

## B.4 Details on Training Schedule

**Implementation Details.** The architectures are trained via backpropagation using the Adam Optimizer[28]. The data is fed to the model in batches of 512 examples for 150 000 steps. We use a modular buffer to sample an important variety of different descriptions in each batch and to impose a ratio of positive samples of 0.1 for each description in each batch.

**Model implementations.** We used the standard implementations of TransformerEncoderLayer and TransformerEncoder from pytorch version 1.7.1, as well as the default LSTM implementation. For initialization, we also use pytorch defaults.

**Hyper-parameter search.** To pick the best set of parameters for each of our eight models, we train them on 18 conditions and select the best models. Note that each condition is run for 3 seeds and best models are selected according to their averaged $F_1$ score on randomly held-out descriptions (15% of the sentences in each category given in Table 1).

**Best models.** Best models obtained thanks to the parameter search are given in Table 2.

| Model | Learning rate | Model hyperparams | | | |
|---|---|---|---|---|---|
| | | hidden size | layer count | head count | param count |
| UT | 1e-4 | 256 | 4 | 8 | 1.3M |
| UT-WA | 1e-5 | 512 | 4 | 8 | 14.0M |
| TFT | 1e-4 | 256 | 4 | 4 | 3.5M |
| TFT-WA | 1e-5 | 512 | 4 | 8 | 20.3M |
| SFT | 1e-4 | 256 | 4 | 4 | 3.5M |
| SFT-WA | 1e-4 | 256 | 2 | 8 | 2.7M |
| LSTM-FLAT | 1e-4 | 512 | 4 | N/A | 15.6M |
| LSTM-FACTORED | 1e-4 | 512 | 4 | N/A | 17.6M |

Table 2: **Hyperparameters.** (for all models)

**Robustness to hyperparameters** For some models, we have observed a lack of robustness to hyperparameters during our search. This translated to models learning to predict all observation-sentence tuples as false since the dataset is imbalanced (the proportion of true samples is 0.1). This behavior was systematically observed with a series of models whose hyperparameters are listed in Table 3. This happens with the biggest models with high learning rates, especially with the -WA variants.

| Model | Learning rate | Model hyperparams | | |
|---|---|---|---|---|
| | | hidden size | layer count | head count |
| UT-WA | 1e-4 | 512 | 4 | 4 |
| UT-WA | 1e-4 | 512 | 4 | 8 |
| SFT | 1e-4 | 512 | 4 | 4 |
| SFT-WA | 1e-4 | 512 | 4 | 8 |
| SFT-WA | 1e-4 | 512 | 2 | 4 |
| SFT-WA | 1e-4 | 512 | 4 | 4 |
| TFT | 1e-4 | 512 | 4 | 4 |
| TFT-WA | 1e-4 | 512 | 4 | 8 |
| TFT-WA | 1e-4 | 512 | 2 | 4 |
| TFT-WA | 1e-4 | 512 | 4 | 4 |

Table 3: **Unstable models.** Models and hyperparameters collapsing into uniform false prediction.

## C  Supplementary Discussion: Formal descriptions of spatio-temporal meanings

The study of spatial and temporal aspects of language has a long history in Artificial Intelligence and linguistics, where researchers have tried to define formally the semantics of such uses of language. For instance, work in temporal logic [2] has tried to create rigorous definitions of various temporal aspects of action reflected in the English language, such as logical operations on time intervals (an action fulfilling itself simultaneously with another, before, or after), non-action events (standing still for one hour), and event causality. These formal approaches have been complemented by work in pragmatics trying to define language user's semantics as relates to spatial and temporal aspects of language. For instance, Tenbrink [43] examines the possible analogies to be made between relationships between objects in the spatial domain and relationships between events in a temporal domain, and concludes empirically that these aspects of language are not isomorphic and have their own specific rules. Within the same perspective, a formal ontology of space is developed in [4], whose complete system can be used to achieve contextualized interpretations of language users' spatial language. Spatial relations in everyday language use are also specified by the perspective used by the speaker; a formal account of this system is given in [44], where the transferability of these representations to temporal relations between events is also studied. These lines of work are of great relevance to our approach, especially the ones involving spatial relationships. We circumvent the problem of reference frames by placing ourselves in an absolute reference system where the x-y directions unambiguously define the concepts of *left*, *right*, *top*, *bottom*; nevertheless these studies would be very useful in a context where the speaker would also be embodied and speak from a different perspective. As for the temporal aspect, these lines of work focus on temporal relations between separate events, which is not the object of our study here; we are concerned about single actions (as opposed to several events) unfolding, in the past or present, over several time steps.

# D Supplementary Results

## D.1 Generalization to new observations from known sentences

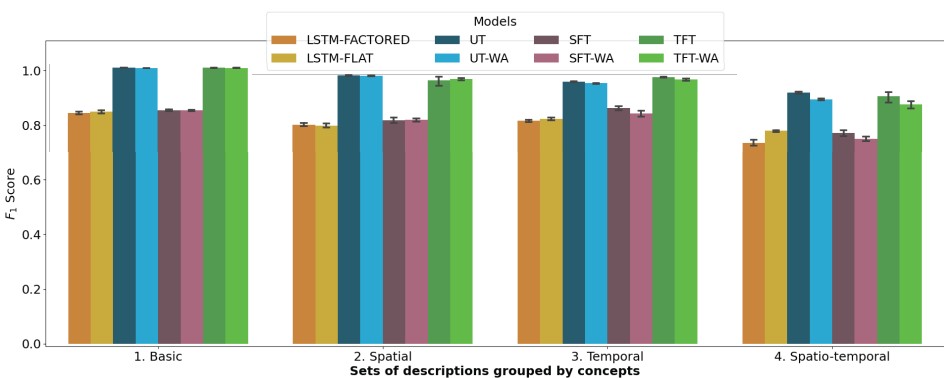

Figure 9: **Generalization to new traces of observations.** F1 scores of all models on the train sentences with new observations. UT and TFT outperform other models on all four categories of meanings.

## D.2 Computing Resources

This work was performed using HPC resources from GENCI-IDRIS (Grant 2020-A0091011996). We used 22k GPU-hours on nvidia-V100 GPUs for the development phase, hyperparameter search, and the main experiments.