# OpenReview forum: "Grounding Spatio-Temporal Language with Transformers"
_NeurIPS.cc/2021/Conference — NeurIPS 2021 Poster_

### Official Review · Reviewer_P8h7 · 2021-07-09

**Rating:** 5
**Confidence:** 4

**Summary:**

This paper proposes a new dataset for reasoning about vision, language, and time. The dataset (Temporal Playground) consists of several objects moving around an image paired with a sentence that may or may not describe the image. The task is to determine whether the description is true or false about the moving image. The data is synthetically generated.

The paper also proposes several architectures to solve this task and dataset, and experiments with several settings including how the inputs are combined and how to represent sentences in the model.

**Limitations And Societal Impact:**

The authors discuss the main limitation (in my opinion) which is the synthetic nature of the dataset.

The societal impact of the work is briefly discussed. I don't have specific suggestions for it, but at the very least I would like a deeper discussion about what this dataset in particular offers over existing ones for real-world applications.

**Main Review:**

Originality: As mentioned in this paper, there are many datasets that describe video captions on real videos (e.g., HowTo100M of Miech et al. 2019), and datasets which are completely synthetically generated based on CLEVR (e.g., CLEVRER of Yi et al. 2020) that focus on image reasoning. I don't understand the distinction between CLEVRER and this proposed dataset, as CLEVRER requires spatial and temporal reasoning, and though I'm not familiar with all of the evaluation splits of the dataset, one could imagine resplitting it to evaluate different kinds of generalization (e.g., unseen sentences/questions). It isn't clear what this proposed dataset adds that existing datasets don't already have. Since the task is fairly specific, I would like to see a more in-depth discussion on the possible applications of this data (and what if offers over existing datasets).

The experiments comparing SFT and TFT could have wider applicability beyond this dataset to other video processing work. I'm not very familiar with existing work in video processing, and not sure if this kind of experiment has been done before.

Quality: The dataset is synthetically generated, so it is not clear how work on this data will extend to natural language (is the "Natural English version" in Figure 1 ever used?). The performance on the dataset is already very impressive, and I would have liked to see more detailed error analysis about what the models are failing on. The motivation behind some of the choices in the dataset (e.g., the number of object types and properties, including interaction between object types, etc.) is not well-stated.

Results showing a small difference between using and not using WA don't seem very meaningful here, because the language is relatively simple and synthetically generated (i.e., the conclusion that language can be summarized into a single vector wouldn't necessarily hold in the real world). It would be interesting to see baselines that provide only one of the two inputs (video or text, not both) to the model.

The claim that "Meanings typically refer to instantaneous actions and rarely consider spatial reference to objects" is not true. There are plenty of challenging grounded instruction-following datasets that require resolving complex spatial relationships between objects and following long, ordered instructions, e.g., Touchdown (Chen et al. 2019).

Clarity: The paper is relatively easy to understand. The intro was fairly low-level near the end (not sure what a "grammar primitive" is in this context). I was confused about what the "agent" was in Section 2.2 -- how was this controlled? I would have preferred to see the equations in lines 177--178 written in standard matrix operation notation rather than a pytorch-like syntax.

Significance: The results on SFT vs. TFT may be applicable to other systems which process video data. Other researchers may use this dataset in addition to similar datasets like CLEVRER. It's not clear to me the main contribution of this work over existing works, and I worry about its limitations as a synthetically generated dataset.

**Time Spent Reviewing:**

1.5

---

> ### Author Response · Authors · 2021-08-10
> **Response to Reviewer P8h7: Details on the envisioned application of the dataset in a broader context, on the extension to natural language, and on comparison with existing language grounding benchmarks**
>
> We thank the reviewer P8h7 for their thorough and constructive review, and for giving us the opportunity to discuss the significance of this work in the wider scope of language grounding and autonomous agents that we have in mind.
>
> *“I would like to see a more in-depth discussion on the possible applications of this data (and what if offers over existing datasets).”*
>
> The main application we envision for trained multimodal truth functions are autonomous embodied goal-conditioned RL agents. Such agents are trained with RL to follow linguistic instructions, and use this learned truth function linking observations and language as the reward function to optimize. The goal-conditioned policy optimizes the learned reward function which is in turn trained in a binary classification framework to match observations and linguistic descriptions. This latter data is given by an oracle. An example of this setting is given in this work [1], and it is very important since it allows to implement autonomous agents that discover, through linguistic feedback and in an incremental manner, the interesting interactions to be had with its environment. In this context, systematic generalization is a primordial requirement for the reward function, because it allows the agent to train on novel, out-of distribution language goals.
>
> Within this framework we are interested in studying which architectures could work for temporally-extended predicates such as ‘shake’. This is not something that we are able to do with the CLEVRER dataset, which focuses on causal reasoning about collision events and not on interaction traces of an embodied agent. Therefore work with CLEVRER cannot be directly extended for RL problems involving an embodied agent.
>
> *“The dataset is synthetically generated, so it is not clear how work on this data will extend to natural language”*
>
> Studying language grounding and linguistic generalization with synthetic language has a long history in the literature. Synthetic language, contrary to natural language, is fully compositional and does not contain exceptions. This is, thus, the ideal setting for studying the compositional abilities of neural networks. Furthermore, as discussed in our conclusion (l360), there exist techniques that enable the application of learning systems trained on synthetic language to natural language settings with varied grammatical structures and synonyms. Use of synthetic language for studying linguistic generalization is standard at least since the seminal SCAN [2] benchmark, and has been of great help in clarifying the debate about linguistic generalization [3] and defining metrics on difficulty of generalization [4]. As far as language grounding is concerned, most benchmarks provide synthetically generated data so as to ease data collection, evaluation and analysis of the proposed methods. Works like BabyAI [5] and gSCAN [6] come to mind.
>
> *“is the "Natural English version" in Figure 1 ever used?”*
>
> The “Natural English version” is given for illustrative purposes only. We clarified the figure caption to make this clearer.
>
> *“Meanings typically refer to instantaneous actions and rarely consider spatial reference to objects" is not true”*
> *“I would like a deeper discussion about what this dataset in particular offers over existing ones for real-world applications.”*
>
> With the following response we hope to answer both these remarks of the reviewer.
>
> Other works in Language Grounding indeed consider settings that involve both spatial and temporal concepts (CLEVR [5], CLEVERER [6], ALFRED [7], BabyAI [5], gSCAN [6], Touchdown [8] etc..). Nevertheless, the types of temporally-extended semantics (actions spanning several time steps, past modality) we consider are not represented in this literature.
>
> First, some of these works (AFLRED, Touchdown for instance) follow the Vision-and Language Navigation paradigm (VLN). This is a very important problem, in particular in terms of language grounding (it involves visual reference to object and places and spatial concepts) but is distinct from learning to ground descriptions of temporally-extended actions of an agent in an environment such as shaking. The fact that those benchmarks use natural language makes them more relevant for industrial applications but less amenable to detailed analysis.
>
> Other works in language grounding are based on synthetic language (CLEVR/CLEVRER, BabyAI, gSCAN), but do not offer all the aspects we consider in our work. Both CLEVR and CLEVRER lack an embodied agent, CLEVRER does not consider spatial relationships between objects, BabyAI does not consider temporally-extended actions (instead focusing on sequential chaining of goals). gSCAN is close in spirit to our work, as they specifically target systematic generalization. Nevertheless their task consists in the (open-loop) generation of a sequence of actions given a single frame of observation from the environment. Thus they do not ground their actions in temporally-extended observations. This also reduces the architectural decisions to be made for processing the observation, which is the object of our study with different architectural variants.
>
> *“I would have preferred to see the equations in lines 177--178 written in standard matrix operation notation rather than a pytorch-like syntax”*
>
> We will additionally add, in the final version of the paper, the matrix notation of one transformer layer as well as their composition.
>
> [1] C. Colas, T. Karch, N. Lair, J.M. Dussoux, C. Moulin-Frier, P. Dominey, P.Y. Oudeyer - Language as a Cognitive Tool to imagine goals in Curiosity-Driven Exploration - Proceedings of Neural Information Processing Systems, 2020
>
> [2] Brenden M. Lake and Marco Baroni. Generalization without systematicity:  On the compositional skills of sequence-to-sequence recurrent networks, Proceedings of the International Conference of Machine Learning, 2018
>
> [3] Dieuwke Hupkes, Verna Dankers, Mathijs Mul, and Elia Bruni. Compositionality decomposed:  how do neural networks generalise? Journal of Artificial Intelligence Research, 2020
>
> [4] Daniel Keysers, Nathanael Scharli, Nathan Scales, Hylke Buisman, Daniel Furrer, Sergii Kashubin, Nikola Momchev, Danila Sinopalnikov, Lukasz Stafiniak, Tibor Tihon, Dmitry Tsarkov, Xiao Wang, Marc van Zee, and Olivier Bousquet.  Measuring compositional generalization:  A comprehensive method on realistic data, Proceedings of the International Conference on Learning Representations, 2020.
>
> [5] Justin  Johnson,  Bharath  Hariharan,  Laurens  van  der  Maaten,  Li  Fei-Fei, C.  Lawrence  Zitnick,  and  Ross  Girshick.   Clevr:   A  diagnostic  dataset  for compositional language and elementary visual reasoning, 2016.
>
> [6] Kexin Yi, Chuang Gan, Yunzhu Li, Pushmeet Kohli, Jiajun Wu, Antonio Torralba, and Joshua B. Tenenbaum.  Clevrer:  Collision events for video representation and reasoning, Proceedings of the International Conference on Learning Representations, 2020.
>
> [7] Mohit Shridhar, Jesse Thomason, Daniel Gordon, Yonatan Bisk, Winson Han, Roozbeh Mottaghi, Luke Zettlemoyer, and Dieter Fox. Alfred:  A benchmark for interpreting grounded instructions for everyday tasks, Proceedings of CVPR, 2020.
>
> [8] Howard  Chen,  Alane  Suhr,  Dipendra  Misra,  Noah  Snavely,  and  Yoav  Artzi. Touchdown:   Natural  language  navigation  and  spatial  reasoning  in  visual street environments, Proceedings of CVPR, 2019.

---

> > ### Comment · Reviewer_P8h7 · 2021-08-15
> > **Reply**
> >
> > Thanks for your response. This clarifies the motivation a bit for me, and I can more easily see what the intent of a dataset like this is for.
> >
> > However, I am still leaning towards rejection. In my opinion, a synthetic dataset with very good baselines intended for use in embodied RL settings where the agent must behave very specific ways during its instruction execution (e.g., must shake an object at some point) is not sufficient for publication. As it stands, from this paper it seems the goal is to encourage benchmarking on this dataset, while the focus should be on the end goal of building an embodied RL system which uses a function learned from data like this as a reward. I understand the authors can clarify their motivation in the paper to focus more on this end goal, and I would suggest doing so, but even with this change only I don't think the contributions are enough for publication.
> >
> > At the very least, I would like to see how this system actually works in practice -- i.e., in an RL setting where it is used to compute reward. Also, this being the end goal of this dataset, it does seem important to consider the role of natural language, because task specifications in the real world will not always fit within the templates defined in this paper. I agree that synthetic data can be useful for debugging, and I also appreciate that the authors are focusing on what seems to be an understudied problem of describing actions across multiple timesteps via abstractions like "shake".

---

> > > ### Author Response · Authors · 2021-08-20
> > > **Justifications on the study of the reward function in isolation and the use of synthetic language**
> > >
> > > We thank the reviewer for their fast response. We are glad that our motivations appear clear now. There are, however, some aspects we would like to develop.
> > >
> > > This paper does not necessarily focus on encouraging benchmarking. Instead, it proposes the first step for the fundamental understanding of the Spatio-temporal language grounding problem and possible methods to solve it.
> > >
> > > We propose a thorough analysis of a truth function. As we already discussed, it is a central aspect of embodied RL agents. We hope that this work can motivate future projects with embodied RL agents manipulating Spatio-temporal language. Such agents would need to embed our reward function architecture to do so. However, we do not believe that including RL experiments would improve this paper, as:
> > >
> > > 1. we think it is well principled to focus on the study of the reward function, and results of RL would depend a lot on the RL machinery used, which is an orthogonal problem;
> > > 2. this would decrease the space for analysis of the current study, and as other reviewers noted, there is value in investigating the reward function properties in isolation. Unfortunately, this is incompatible with adding an RL experiment.
> > >
> > > Finally, using synthetic language is not for debugging, but for fundamental understanding. Current work in RL cannot yet tackle the immense variety of the language we speak. Hence, there are many sorts of ''natural language'' that depend a lot on applications. As such, it makes more sense to focus on this in a paper dedicated to a specific application with an associated kind of ''natural language''

---

### Official Review · Reviewer_Zjb3 · 2021-07-16

**Rating:** 7
**Confidence:** 3

**Summary:**

The study presents a new synthetic dataset for the Embodied Language Grounding task, which specifically targets spatio-temporal aspects of language. The authors further test a number of transformer-based models on the task, identifying a number of architectural properties that are likely to be conducive to specific types of generalization (three such types were considered).

The paper strikes me as a well-rounded and a high-quality contribution.

**Limitations And Societal Impact:**

Broader impact and limitations are discussed in sufficient depth.

**Main Review:**

## Strengths

- The paper addresses a highly relevant problem
- The paper is sufficiently novel and original
- The paper is well written and is a pleasure to read
- Experiments are thorough and sufficient to support the claims being made

## Weaknesses

- The paper focuses on a simulated dataset, which limits direct practical applications
- The models seem to achieve near-ceiling performance in a number of conditions

Overall, I enjoyed reading this paper and I believe that it is a strong contribution that may aid in the challenging and important task of Language Grounding.

The main concern I have is that the proposed dataset may be too easy to be a "driving force" behind architecture development. In the most challenging conditions, at least some of the models considered in the paper already seem to reach near-ceiling performance.

I believe that expanding the artificial language so as to be able to present an even more trying challenge would have made the paper noticeably stronger. Ideally, a simulated dataset should offer a continuum of tasks (from extremely easy to extremely difficult).

This concern is not too crucial, since there are conditions that are not yet solved, and hence there is room for improvement in model development. Hopefully, by the time the dataset is solved completely, its modified version will be developed and released.

Regardless of the concern voiced above, I believe that the experimental results will be interesting for a large number of researchers, and that the dataset, although not always challenging enough, may still be a good starting point for development of grounded linguistic agents.

## Typos

362 on -> own


**Time Spent Reviewing:**

3

---

> ### Author Response · Authors · 2021-08-10
> **Response to Reviewer Zjb3**
>
> We are thankful to Reviewer Zjb3 for their kind words and provide extra information about our motivations in our response to reviewer xTkz that might answer their slight concern. We would be happy to answer any of their questions that may arise from reading other reviews.

---

### Official Review · Reviewer_3aA5 · 2021-07-16

**Rating:** 6
**Confidence:** 4

**Summary:**

This paper provides a study of spatio temporal language grounding. It introduces a new environment called the Temporal Playground that generates synthetic language descriptions of the traces of the actions performed by an embodied agent. Although the relative search space of synthetic language is limited, it provides a set of experiments to demonstrate model generalization to aspects like unseen word combinations. A study of variations on transformer architectures for this task is also provided.

**Ethical Concerns:**

I have no ethical concerns for this work.

**Limitations And Societal Impact:**

Though plenty of downstream application based concerns exist for such embodied agents, at its current stage - transfer to them is not yet possible. There is thus limited societal impact.

**Main Review:**

Pros:

The motivation is quite clear, lots of vision+language sequential decision making problems but not many have really focused on the spatio-temporal aspects of the language being used.

The paper is clearly written and I could follow along at all steps, experiment and architecture choices are clearly defined.

The study of providing temporal bias first vs spatial bias first in the transformer architecture (as opposed to just the unstructured transformer) is potentially valuable to other domains involving sequential decision making.


Cons/Questions:

The selection of which predicates involve temporal concepts and which don't seem rather arbitrary. A notion of time units in terms of steps could perhaps help here. Else one could make the argument that an action such as "grasp" is also a temporal concept in addition to being a spatial one in all cases (and not only when presented as a past tense predicate).

Many of the conclusions regarding the linguistic components being tested (e.g. lines 230-231) are likely to be complicated by factors such as the size of the synthetic vocabulary and the number of novel descriptions. Even within the limited vocab used, an ablation showing performance of the models on the experiments such as within Section 3.1 as a function of the size of the synthetic vocab used would be critical towards validating such claims.

Similarly to the above concern, the tests of systematic generalization on withheld combinations of words only tests on a one to three forbidden training samples. A much more useful experiment here would be to ablate this as a function of the number of forbidden training word combination (1, 5, 10...). As it is, it is difficult to tell whether the models are truly generalizing to these unseen combinations or simply finding proxy features in the surrounding text.

(Minor) A zeroshot eval on description traces from "real language" (something like the Room2Room dataset https://bringmeaspoon.org/ or existing prodedural text understanding datasets) would be very useful as a sanity check. I'm not asking for super high performance, just that it would better ground a reader's judgement on how the synthetic eval numbers match up when evaluated on a larger language observation space.

**Time Spent Reviewing:**

3

---

> ### Author Response · Authors · 2021-08-10
> **Response to Reviewer 3aA5: Clarification of the definition of temporal predicates and new model evaluations**
>
> We thank reviewer 3aA5 for their thorough review and for suggesting a variety of valuable ablation studies that we are happy to discuss here.
>
> Before diving into the ablations studies, let’s first answer a question about our setup:
>
> *‘“The selection of which predicates involve temporal concepts and which don't seem rather arbitrary. A notion of time units in terms of steps could perhaps help here.’’*
>
> We decided to differentiate temporal from instantaneous predicates by simply stating that temporally-extended predicates are the ones whose truth value require several time steps of observations to be inferred. Grasp can be decided from only a single frame of a trajectory whereas Grow and Shake cannot. Growing an object typically spans over 5 to 10 time units depending on the growing rate. Similarly, shaking an object requires at least 3 oscillations and thus spans over at least 6 time units. Extending our dataset with a richer set of temporal predicates is an interesting avenue for future work. In this perspective, we agree that a classification of predicates in terms of time duration is a great idea.
>
> Reviewer 3aA5 suggests two interesting ablation studies with a variation of the vocabulary size used for training and a variation of the number of words used in forbidden combinations used for testing. Those two analyses are indeed highly valuable but require creating new datasets and retraining all our baselines. Due to the high computational cost of this procedure, we are not able to conduct such experiments during the period of this rebuttal. However, in order to answer reviewer 3aA5’s concern that it is difficult ‘’to tell whether the models are truly generalizing to these unseen combinations or simply finding proxy features in the surrounding text’’, we performed two control experiments that do not imply retraining all baselines. More specifically we tested our classification models in two hard negative examples scenarios in order get more insight into the model’s ability to understand compositional language. We defined to set of tests:
>
> - 1. Hard trajectories where negative pairs (trajectory, description) are selected so that the trajectories contain the object or the actions described by the descriptions;
> - 2. Hard descriptions where negative pairs (trajectory, description) are selected so that the trajectories match with the descriptions but the descriptions are mutated. Mutations are performed by randomly shuffling the order of words in the ground truth descriptions.
>
> In the first setting with hard trajectories, we observe a slight decrease of performances on all 5 categories (drop is less than 5%), demonstrating that the models do in fact represent the meaning of the sentence and not simply the presence or absence of a particular object or predicate.
>
> In the second setting with hard descriptions, we observe an important drop of performances (around 0.3 points across models) on the first two categories of systematic generalization (new object-attribute and new predicate-object combinations). Conversely, the three other categories of systematic generalization show similar performances as in the random negative setting. This result can be explained by the fact that, for the two first categories, word ordering does not impact the meaning of descriptions. Therefore, it is sufficient for the models to perform unit concept recognition in order to get high f1 scores in the train, and when tested on randomly re-shuffled sentences this partial permutation invariance prevents the model from considering these sentences as negative. This is not the case in the harder systematic splits 3 to 5, where the ordering of words does matter for deciding the sentence in the train set (think of the position of the token ‘was’, or the order of words in spatial relations). To summarize, the models do seem to recognize more than simple concept units, even if there is a partial effect of individual, independent recognition in the first and second systematic split.

---

> > ### Comment · Reviewer_3aA5 · 2021-09-01
> > **Reply**
> >
> > Thanks for the response. The clarification on the time units and how they map to frames in the environment is quite useful and I'd encourage the authors to add more on this to the main text. The control experiments serve to ease some of my concerns regarding generalization. I will not increase my score, however, as my primary concern (and other reviewers too looks like) regarding the utility of the (synthetic) task with respect to the underlying motivation is unclear. There's nothing wrong with having a task based on synthetic data but without any notion of scaling (e.g. on vocab size) or on how models do with closely related "real language" tasks - it is difficult to fully substantiate the claims the authors have made.

---

### Official Review · Reviewer_xTkz · 2021-07-17

**Rating:** 4
**Confidence:** 4

**Summary:**

In this paper, the authors introduce a spatio-temporal language grounding task. In their setting, the agent interacts with the surrounding objects over time. Once the episode is over, an oracle provides exhaustive feedback in a synthetic language about everything that has happened. They use the transformer model to solve the task. They also test the generalization ability of their models, including the generalization to randomly held-out sentences and the generalization to grammar primitives.

**Limitations And Societal Impact:**

The motivation is not clear and the claimed contributions are minor.
The authors claim that they "study the grounding of new spatio-temporal concepts enabling agents to ground time extended predicates with complex spatio-temporal references to objects and understand both present and past tenses". This sounds confusing. As I know, there are some language grounding papers that involve both spatial and temporal concepts, such as

-- ALFRED A Benchmark for Interpreting Grounded Instructions for Everyday Tasks
-- Are We There Yet? Learning to Localize in Embodied Instruction Following
-- FollowNet: Robot Navigation by Following Natural Language Directions with Deep Reinforcement Learning
-- Vision-and-Language Navigation: Interpreting visually-grounded navigation instructions in real environments
-- IQA: Visual Question Answering in Interactive Environments
-- PIGLeT: Language Grounding Through Neuro-Symbolic Interaction in a 3D World

The Problem Definition section and Figure 1 are confusing.
How did you get the synthetic linguistic descriptions? How does the oracle provide exhaustive feedback?
The tasks are very simple. It seems that the linguistic descriptions are script generated and are not diverse enough.
How did you control that the generated trajectories make sense? What if the agent repeats one action or alternatively executes two actions? It is unclear how do you get good trajectories and how does the oracle work.

The writing needs to be improved.



**Main Review:**

It is good that the paper considers the spatial-temporal language descriptions in the language grounding tasks. However, the motivation is not clear and the claimed contributions are minor.

The authors claim that they "study the grounding of new spatio-temporal concepts enabling agents to ground time extended predicates with complex spatio-temporal references to objects and understand both present and past tenses". This sounds confusing. As I know, there are some language grounding papers that involve both spatial and temporal concepts. See detailed limitations below.

The task definition is not clear. What is the difference between the proposed Embodied Language Grounding task from existing works? What is the advantage of the new task? Why is it more challenging than existing language Grounding tasks?

The details of how to get the trajectories and languages are missing. See detailed limitations below.

Furthermore, if you can generate linguistic descriptions given trajectories by the oracle function, it seems you can also get the trajectories given the linguistic descriptions using scripts, especially that the environment is very simple and not diverse. It seems not hard to hand-code the action sequences given some linguistic descriptions. Then why do you still need to train a model for language grounding? If it is impossible to get the action sequences by hand-coding, could you explain why?

It is good that the authors show the generalization experiments. Is it because of the Transformer model? If so, what's the novelty of your method? The good generalization ability of Transformer has been shown in many papers.









**Time Spent Reviewing:**

5 hours

---

> ### Author Response · Authors · 2021-08-10
> **Response to Reviewer xTkz: Position in the language grounding literature, on the appropriateness of training reward functions, and experimental details**
>
> We thank reviewer xTkz for their constructive feedback and hope to clarify our motivations and methods through this discussion. Other works in Language Grounding indeed consider settings that involve both spatial and temporal concepts (CLEVR [1], CLEVERER [2], ALFRED [3], BABYAI [4], gSCAN [5], etc..). Benchmarks often involve navigation or object manipulation with instructions that target abstract instantaneous configurations. In this context, behaviors do require systems that analyze temporal and spatial relations in a series of observations. However, the descriptions aligned with those trajectories only include some limited aspects of Spatio-temporal language. In particular, they do not consider explicitly time-extended predicates that describe behaviors spanning over multiple time-steps, past tense predicates, and complex spatio-temporal references to objects based on their past configurations. For instance, in ALFRED, goal states are expressed through the instantaneous state of certain objects (a cooked potato on a plate). In gSCAN, the model is asked to produce a sequence of action commands using an instantaneous observation; we provide the whole trace of the episode as observation. Similarly to ALFRED, BABYAI only considers sequences of actions as the temporal aspect, and not temporally-extended actions. In this paper, we propose to study a variety of temporal predicates through systematic experiments that investigate the generalization of several learning architectures. Moreover, in opposition to recent work on spatio-temporal reasoning (CLEVERER), we carefully analyze the systematic generalization capabilities of our system in language space. For this reason, our new task differs from other existing Language Grounding problems. To sum up, our motivation is to provide a controlled dataset with a synthetic language as well as a first set of baselines that can inspire future work in this direction.
>
> *‘’It seems not hard to hand-code the action sequences given some linguistic descriptions. Then why do you still need to train a model for language grounding? If it is impossible to get the action sequences by hand-coding, could you explain why?’’*
>
> It is indeed straightforward to hand-code the action sequences that correspond to the linguistic descriptions of our setup. We implemented such a bot to generate the dataset we use for training. This, however, does not reduce the interest in proposing algorithms for learning truth function detecting when a description matches with a certain behavior. In fact, many machine learning benchmarks are based on oracle procedures and synthetically generated data (in fact, all the ones mentioned previously: CLEVR, CLEVERER, ALFRED, BABYAI, gSCAN). This is useful from a fundamental research perspective, since synthetic data allows us to better understand the behavior of models and the influence of architectural inductive biases in a controlled setup. Thus we are not interested in solving an engineering problem, but in studying from a developmental perspective what kind of meanings can be grounded and by which architectures.
>
> Moreover, truth functions model a specific, yet crucial and general, type of meaning in Language Grounding. Within a Reinforcement Learning perspective, where agents are trained to reach language goals, it can be used as a reward signal for training [1,2]. In this direction, designing agents that can convert Spatio-temporal descriptions provided by a Social Partner into rewards for training, would enable them to self-train policies that reproduce the trajectories they represent. This paves the way towards machines that learn rich behaviors from social interactions with (artificial) caregivers.
>
> *‘’It is good that the authors show the generalization experiments. Is it because of the Transformer model? If so, what's the novelty of your method? ‘’*
>
> We would like to ask the reviewer to clarify their question here, as we are not sure what they meant to ask. Nevertheless, as concerns architectures, the aim of our work was to provide a  systematic study of different architectural inductive biases in our spatio-temporal language grounding task. The finding that object identity plays an important role in generalization is a conclusion that can guide further work in spatio-temporal language grounding, including in extended environments with more sophisticated language and time-extended predicates, and richer observations. Furthermore, the fact that the UT architecture, which is the one with the least architectural inductive biases, seems to be the most robust is an important basis for extension, for instance to more modalities.
>
> *“The details of how to get the trajectories and languages are missing. See detailed limitations below.”*
>
> We agree with reviewer xTkz that we give insufficient details on trajectory generation, and we will provide a more exhaustive description in the camera-ready version of the manuscript after this discussion. We implement a bot which executes diverse behaviors such as going to objects, grasping  them, moving them around, and bringing supplies to plants and animals to make them grow. We also implement a logical system that is able to generate all sentences that are valid according to a grammar and true of the generated observation. This allows us to provide exhaustive linguistic feedback about what has happened in the episode. (This can be seen as an exhaustive captioning system in a simplified language for the environment). Since the system also allows us to generate all valid descriptions for the language, we can attribute, for each sentence of the grammar, a truth value given an episode.
>
> [1] Justin  Johnson,  Bharath  Hariharan,  Laurens  van  der  Maaten,  Li  Fei-Fei, C.  Lawrence  Zitnick,  and  Ross  Girshick.   Clevr:   A  diagnostic  dataset  for compositional language and elementary visual reasoning, 2016.
>
> [2] Kexin Yi, Chuang Gan, Yunzhu Li, Pushmeet Kohli, Jiajun Wu, Antonio Torralba, and Joshua B. Tenenbaum.  Clevrer:  Collision events for video representation and reasoning, Proceedings of the International Conference on Learning Representations, 2020.
>
> [3] Mohit Shridhar, Jesse Thomason, Daniel Gordon, Yonatan Bisk, Winson Han, Roozbeh Mottaghi, Luke Zettlemoyer, and Dieter Fox. Alfred:  A benchmark for interpreting grounded instructions for everyday tasks, Proceedings of CVPR, 2020.
>
> [4] Maxime Chevalier-Boisvert, Dzmitry Bahdanau, Salem Lahlou, Lucas Willems, Chitwan Saharia, Thien Huu Nguyen, and Yoshua Bengio.  Babyai:  A platform to study the sample efficiency of grounded language learning, Proceedings of the International Conference on Learning Representations, 2019.
>
> [5] Laura Ruis, Jacob Andreas, Marco Baroni, Diane Bouchacourt, and Brenden M. Lake.  A benchmark for systematic generalization in grounded language understanding, Proceedings of Neural Information Processing Systems, 2020.

---

### Decision · Program_Chairs · 2021-09-28

**Decision:**

Accept (Poster)

**Comment:**

This paper proposes the task of predicting truthfulness of a summary statement given an agent’s behavioral trace in an environment, as a way of measuring a model’s understanding of spatio-temporal concepts. The reviewers had differing opinions about the paper — while they all agreed that the task of spatio-temporal grounding is interesting and impactful to solve, there were several concerns around the motivation of the paper being unclear, the synthetic nature of the task and comparison to existing benchmarks like ALFRED, Room2Room, etc. The author response addressed some questions around the motivation, but the question of how this (synthetic) benchmark would connect to more realistic settings remains unclear. One of the reviewers even suggested re-using data from slightly less synthetic tasks like ALFRED as a step towards demonstrating this connection. On reading the paper myself, I think that the paper has good ideas, but feels too preliminary and in the current form, leaves unanswered important questions such as the comparison with existing datasets and how developing models on this may correlate with real-world tasks (for example, the TFT models are almost at 100% F1 score which begs the question of whether the task is too easy to test all aspects of spatio-temporal grounding).

**Consistency Experiment:**

NeurIPS has a long history of experimentation. In 2014, NeurIPS ran an experiment in which 10% of submissions were reviewed by two independent committees to quantify the randomness in the review process. This year, we repeated a variant of this experiment to see how the quality of the review process has changed over time.  This paper was part of the experiment and was therefore assigned to two committees (consisting of reviewers, an Area Chair, and a Senior Area Chair) that reached independent decisions.  If both committees made the same recommendation, this recommendation was followed. If a single committee recommended acceptance, the paper was accepted (with the exception of a few cases in which the other committee identified what we considered a fatal flaw, e.g., an error in a key result).

This copy’s committee reached the following decision: **Reject**

The other committee assigned to the paper recommended **Accept (Poster)**.  You can find the other set of reviews, along with any follow up discussion with the authors here:
https://openreview.net/forum?id=ZQQqo8H1qjC